# Suberoylanilide Hydroxamic Acid (SAHA) Is a Driver Molecule of Neuroplasticity: Implication for Neurological Diseases

**DOI:** 10.3390/biom13091301

**Published:** 2023-08-24

**Authors:** Lucia Verrillo, Rosita Di Palma, Alberto de Bellis, Denise Drongitis, Maria Giuseppina Miano

**Affiliations:** 1Institute of Genetics and Biophysics Adriano Buzzati-Traverso, CNR, 80131 Naples, Italy; lucia.verrillo@igb.cnr.it (L.V.); rosita.dipalma@unina.it (R.D.P.); 2A.O.R.N. S. Anna and S. Sebastiano Hospital, Division of Neurosurgery, 81100 Caserta, Italy; albertodebellis@hotmail.com; 3Maria Rosaria Maglione Foundation Onlus, 80122 Naples, Italy

**Keywords:** neuro-epigenetics, deacetylase inhibitor, protein acetylation, *in vitro* and *in vivo* treatments, disease-modifying therapies

## Abstract

Neuroplasticity is a crucial property of the central nervous system to change its activity in response to intrinsic or extrinsic stimuli. This is mainly achieved through the promotion of changes in the epigenome. One of the epi-drivers priming this process is suberoylanilide hydroxamic acid (SAHA or Vorinostat), a pan-histone deacetylase inhibitor that modulates and promotes neuroplasticity in healthy and disease conditions. Knowledge of the specific molecular changes induced by this epidrug is an important area of neuro-epigenetics for the identification of new compounds to treat cognition impairment and/or epilepsy. In this review, we summarize the findings obtained in cellular and animal models of various brain disorders, highlighting the multiple mechanisms activated by SAHA, such as improvement of memory, learning and behavior, and correction of faulty neuronal functioning. Supporting this evidence, *in vitro* and *in vivo* data underline how SAHA positively regulates the expression of neuronal genes and microtubule dynamics, induces neurite outgrowth and spine density, and enhances synaptic transmission and potentiation. In particular, we outline studies regarding neurodevelopmental disorders with pharmaco-resistant seizures and/or severe cognitive impairment that to date lack effective drug treatments in which SAHA could ameliorate defective neuroplasticity.

## 1. Introduction

Neuroplasticity is a fundamental capacity of neurons to reorganize themselves in response to new information and/or sensory stimulation changing connections in the neuronal network. This complex process occurs most profoundly in the first few years of life as neurons grow very rapidly and send out multiple branches, ultimately forming many connections. Defects in this process may compromise brain development and functioning, as observed in several neurodevelopmental disorders (NDDs).

In the last decade, many studies have been aimed at identifying neuroactive molecules capable of exerting a positive effect on the ability of neurons to reorganize the synaptic network [1]. This branch of research is considered a gold mine in drug discovery applicable to a huge spectrum of neurological disorders with neuroplasticity impairment, including NDDs, psychiatric and neurodegenerative disorders. This review will illustrate how neuroplasticity defects might be productively targeted by suberoylanilide hydroxamic acid (SAHA) or Vorinostat, a clinically approved pan-histone deacetylase inhibitor known to impact various cellular functions by protein acetylation and epigenetic modulation of gene transcription (Figure 1) [2].

Originally approved by the Food and Drug Administration (FDA) for the treatment of T-cell lymphoma [3] and permeable to the blood-brain barrier [4], SAHA was reported as a promising therapeutic molecule for several neurological disorders that still lack effective treatments [5,6]. Remarkably, most of the disease genes involved in these pathologies encode splicing factors and structural proteins implicated in neurogenesis and synaptogenesis. Consequently, mutations in these genes cause aberrant functional processes that can lead to the onset and exacerbation of clinical symptoms, including intellectual disability (ID) and epilepsy severity [7,8]. SAHA is a pan-HDAC inhibitor that inhibits both Class I and Class II HDACs. It belongs to a wide group of HDACI that vary in structures, target selectivity, and biological activities. In recent years, their HDAC inhibitor activity has been exploited as antiproliferative and proapoptotic agents in the treatments of cancer and as neuroprotective and neurotrophic molecules in non-malignant conditions of the nervous system [2,6,9,10,11,12]. This dual activity mirrors the lack of molecular specificity and indicates that the pharmacological effects of SAHA and other HDACIs are based on a wide range of mechanisms of action—a feature common to several drugs—whose identification requires further studies. Of course, given the global effects of SAHA and other HDACIs, it is fundamental to distinguish benefits and detrimental effects, both *in vitro* and *in vivo*. Several studies highlight that, similarly to other HDACIs, the therapeutic outcome of SAHA—effect, efficacy, effectiveness, and benefit—may vary depending on the dosage used and the biological context analyzed (e.g., cell type, tissue specificity, developmental window, etc.) [2,5,13,14,15,16,17,18]. Although future research may elucidate the downstream effects of SAHA, there is a clear trend toward its applicability with new drug repurposing opportunities because (i) it has a good blood-brain barrier permeability [4]; (ii) It does not lead to massive transcriptional changes; and (iii) compared to other HDACi, it is active at nanomolar concentrations in a variety of cellular and animal models [5,6,15,16,17,18]. Future multi-omic studies in *in vivo* models coupled with fast transcriptome and proteome screenings may help to figure out the pharmacology response of SAHA to distinguish at molecular levels the advantages and disadvantages of SAHA, also compared to other HDAC inhibitors, allowing the adoption of personalized applicability of HDAC activity inhibition. Although pharmacogenomic studies for HDAC inhibitors are still scarce, genotype-directed dosing could improve pharmacotherapy of SAHA, reducing toxicity risk or suboptimal dosage. In view of more in-depth studies on the pharmacology response of SAHA, the main point of this review is to highlight the neuroplasticity property of SAHA that could be tested in animal models of neurological disorders that have not been tested to date, giving an overview of cellular and animal models tested, functional processes analyzed, and molecular pathways sensitive to this epi-treatment. In particular, this article aims to review studies on the broad capacity of SAHA in governing neuroplasticity, providing a more complete picture of its multiple activities—at both molecular and cellular levels—such as neuronal maturation, activation of autophagy, microtubule remodeling, and neurospecific splicing modulation (Figure 2) [11,13,14,19,20,21,22,23,24,25,26]. Therefore, we discuss evidence supporting that SAHA-sensitive processes are highly interconnected mechanisms that come together in common neuronal functions. Understanding these multiple activities of SAHA may be helpful to refine preclinical and clinical strategies and potentially accelerate drug repurposing to counteract neurological defects. Finally, by virtue of its ability to stimulate neuritic outgrowth, the promising applicability of SAHA in the treatments of several NDDs is presented and discussed.

## 2. The Role of SAHA as a Neuroactive Compound

Neuroprotective effects refer to the ability of a compound to protect neurons from damage or degeneration, thereby preserving their structure and function. SAHA has been demonstrated to have the potential to exert such neuroprotective effects, which could be beneficial in preventing cognitive decline and maintaining cognitive function. Broader assessments of *in vivo* and *in vitro* treatments have shown that this activity has multiple downstream effects and is not restricted to a particular signaling pathway (Table 1).

Additionally, SAHA has shown neuritogenic effects, meaning it can promote the growth and development of neurites. Neurites are essential for establishing neural connections and facilitating proper communication between neurons. By enhancing neurite growth, SAHA could potentially aid in the restoration of neuronal connectivity, which may be disrupted in cognitive impairment and epilepsy.

By enhancing neurite growth, SAHA could potentially aid in the restoration of neuronal connectivity, which may be disrupted in cognitive impairment and epilepsy. The combination of both neuroprotective and neuritogenic effects of SAHA strengthens its potential for drug repurposing in the treatment of cognition impairment and epilepsy. This section discusses the evidence supporting this feature, emphasizing the potential action of SAHA on neuronal maturation and network development and the importance of analyzing these effects in preclinical models (Figure 3).

### 2.1. Neuronal Maturation and Plasticity

Studies performed by others and by us have shown the ability of SAHA to induce and accelerate neuronal differentiation of embryonic mouse neural stem cells [13,14]. Morphological and molecular data indicate that SAHA-treated ESCs are committed toward neural differentiation with a dose-dependent effect that is cytotoxic at higher doses and pro-proliferative at lower concentrations [13]. Furthermore, it has been shown that SAHA promotes a significant increase in mean neurite length, number of neurites/cell, and neurite length/cell via activation of the ERK pathway [11]. This activity is extremely interesting because in some instances ERK activation may lead to detrimental effects and contribute to neurodegeneration; in other cases, it fosters survival and differentiation or counteracts pathogenic effects of mutations in Huntington’s disease models (HD). Indeed, several studies have reported the dual role of ERK1/2 signaling, as a double-edged sword [34], that in response to specific signals may lead to beneficial effects on long-term and short-term memory formation and neuronal protective activity in Huntington’s disease models (HD) [35,36] and to detrimental effects that induce neurotoxicity [37]. This double role of ERK1/2 signaling may reflect differences in the functional response to SAHA, depending on target cells and tissue, as well as the dosage used [38,39].

Supporting this evidence of SAHA as a proneuronal molecule, we previously showed the activity of this epidrug counteracting the defective neuronal differentiation and maturation in animal and cellular models ablated for aristaless homeobox-related gene (*ARX*), a well-conserved disease gene involved in multiple NDDs [14]. Specifically, SAHA is able to recover KDM5C-H3K4me3 signaling and improve neuronal differentiation both in Arx-KO murine ES-derived neurons and ARX/alr-1-deficient *Caenorhabditis elegans* animals. Behavioral phenotype analysis of alr-1-KO worm mutants revealed that SAHA rescued defective touch response, a behavioral defect affecting mechanosensory neurons [14]. In line with the rationale that SAHA forces gene transcription, our study shows that this epi-molecule corrects the decrease of *Kdm5C*, a gene encoding a histone demethylase involved in NDD and also counteracts the defective transcription of *Bdnf* II and IV isoforms [14] (Figure 3A). Consistent with these findings, it has been shown that SAHA stimulates the expression of BDNF in cultured rat neurons [28] and controls BDNF release in different brain cell populations contributing to memory formation [29]. Furthermore, several investigations proved the association of SAHA with long-lasting chromatin modifications regulating the expression levels of plasticity-related genes or involved in signaling networks that regulate neuronal morphogenesis [30,31,32]. Noteworthy, in human iPSC-derived neurons and mouse embryonic cortical cells, SAHA upregulates—in a dose-dependent manner—the expression of evolutionarily conserved gene networks with key roles in synaptic maintenance and function [40]. Further studies are required to establish in detail the molecular benefits, as well as to balance the possible installation of inappropriate histone modifications.

### 2.2. Activation of Autophagy

Emerging evidence suggests that neuronal autophagy regulates the refinement of the polarized morphology of neurons (e.g., axonal outgrowth, dendritic complexity, and spine pruning) and the formation and maintenance of synapses. Mechanistically, it is a complex catabolic process that delivers cytosolic components to lysosomes for degradation. SAHA has been shown to influence the autophagy pathway at various points and to protect the aged brain against plasticity impairment. This is what emerges from two recent studies showing that cognitive dysfunction induced by the anesthetic sevoflurane or insulin resistance can be counteracted by SAHA, through the expression control of autophagy-related gene markers, such as LC3 and P62 [19,21]. As for the mechanism of action, previous studies carried out in MEF cells demonstrated that SAHA induces autophagy by activating the ULK1 complex, which is the most upstream component in the core autophagy pathway, and by suppressing the mammalian target of rapamycin (mTOR), which in turn inhibits autophagy induction [27] (Figure 3B). While the significance of autophagy in neurodegenerative diseases has been extensively investigated, we are only beginning to understand its functioning during the early stages of brain development [41,42]. Moreover, several studies have shown that perturbations affecting autophagy may cause NDDs. For example, mutations in genes encoding regulatory autophagic proteins have been detected in ASD patients characterized by elevated mTOR activity (mTORopathy) and defective autophagy [42]. Elevated mTOR signaling activity has been also detected in patients with fragile X syndrome [43] and in congenital forms of epileptogenesis and cortical malformations [44]. Given the activity of SAHA as an mTOR inhibitor, this epidrug might be a promising therapy to correct neuronal autophagy in mTOR-related NDDs. However, further studies are required to identify the autophagy-related pathways involved in neurodevelopmental processes in order to identify an appropriate time window for SAHA administration.

### 2.3. Microtubule Organization

Microtubules (MTs) are cylindrical cytoskeletal structures constituted by polymers of αβ-tubulin that play essential roles in many essential mammalian cell functions, such as cytoskeletal support and transportation of intracellular cargo, thus regulating neuroplasticity and synaptogenesis. The dynamic changes regulating MT structure and stability are mediated by acetylation of K40 in α-tubulin, a posttranslational modification, which marks the luminal surface of microtubules and is fundamental for microtubule stabilization and vesicle transportation. The dynamics between acetylation/deacetylation of α-tubulin is mediated by the action of acetyltransferases and deacetylases, respectively. Because of its HDACi activity, SAHA induces hyperacetylation of α-tubulin, facilitating sliding between filaments and making MTs more mechanically stable [11,22] (Figure 3C). A defective level in the α-tubulin acetylation has been detected in animal models for different neurological disorders, including cortical malformations [45], Rett syndrome [46], and various neurodegenerative pathologies [47]. For some of these pathologies, treatments with other HDAC inhibitors rescue the defective α-tubulin acetylation and thus correct a number of MT abnormalities, such as the defective axonal transport observed in Charcot–Marie–Tooth primary neurons [48] and the aberrant BDNF vesicle trafficking observed in Rett patient fibroblasts [46]. Taking this evidence into account, the applicability of SAHA to counteract cytoskeleton malfunctioning in NDDs seems to be very promising to be addressed.

Related to the acetylation/deacetylation effect of SAHA, Luo et al. (2022) underlined a new mechanism of action preventing axonal damage and neurological dysfunction in a rat model of subarachnoid hemorrhage (SAH). This SAHA-induced response is mediated by heat shock protein (HSP) co-inducer activity via the acetylation of HSP70, which in turn induces the degradation of TDP-43 aggregates [33].

### 2.4. Regulation of Alternative Splicing Switch

Alternative splicing (AS) is a critical process of posttranscriptional gene expression that diversifies and expands the proteome and increases the functional diversity of molecules implicated in mammalian brain development and plasticity [49,50]. Characterized by high flexibility, AS is a complex mechanism controlled via RNA-binding proteins (RBPs) forming a megadalton complex named spliceosome [49]. Recent studies highlight that RNA splicing and histone modifications are two related processes, acting via the interaction between HDACs and the spliceosome complex (Figure 3D).

In relation to the levels of histone acetylation, the opening and closing states of chromatin have the ability to influence exon inclusion and exclusion by recruiting directly splicing factors to sites of specific exons [24]. Indeed, histone hyperacetylation, which creates a more open chromatin structure, leads to a faster elongation rate phase allowing for the recruitment of splicing factors to a strong splicing site, resulting in exon skipping. Conversely, a slower elongation rate can recruit splicing factors to weak upstream splice sites, resulting in exon inclusion [24]. In these frameworks, splicing defects can be overcome by manipulating HDACs. In addition to that, a recent study proved that SAHA is also capable to counteract splicing abnormalities by rescuing RNA foci, as demonstrated in a mouse model of myotonic dystrophy type 1 (DM1) [23]. DM1 is the most common form of adult neuromuscular dystrophy caused by the abnormal expansion of CTG repeats in the 3′ untranslated region of the dystrophia myotonic protein kinase (*DMPK*) gene. These expanded repeats lead to the formation of hairpin structures affecting the *DMPK* mRNA (GC-rich foci), which in turn causes the sequestration of splicing regulators such as MBLN1, resulting in aberrant alternative splicing of a variety of mRNAs [16]. The authors proved that SAHA is able to reduce RNA foci formation.

More interestingly, the daily intraperitoneal injections of SAHA in DM1 mice and the daily administration in DM1 patient-derived myoblasts counteract the aberrant splicing affecting myotonia genes (e.g., chloride channel genes) [23]. Moreover, Nakano et al. [25] proved the efficacy of SAHA to counteract aberrant gene expression caused by the defective splicing of the RE1 silencing transcription factor (REST) gene. This gene encodes a transcriptional repressor that forms a protein complex with HDAC1 and HDAC2. Since HDACs are critical for REST-dependent gene expression, the authors demonstrated that the subcutaneous injection of SAHA in *Rest*^+/ΔEx4^ mice induces histone hyperacetylation, rescues defective gene expression of REST target genes, and subsequently induces improvement of cochlear defects [25].

## 3. SAHA in Neurodevelopmental and Psychiatric Diseases: Current Evidence

This section discusses the available evidence in relation to the therapeutic implications of SAHA in a selected number of neurological diseases, affecting the early phases of brain development and characterized by severe cognition impairment and/or pharmaco-resistant epilepsy. Depending on the disease context, studies carried out in animal and cellular models agree that SAHA exerts neuroprotective and neuritogenic effects (Figure 4 and Figure 5). Given SAHA’s beneficial activities, these studies give strength to its repurposing as a promising compound to counteract cognition impairment and epilepsy severity.

### 3.1. Fragile X Syndrome

Fragile X syndrome (FXS) is the most frequent inherited form of intellectual disability (ID) and autism spectrum disorder (ASD), caused by the silencing of the FMR1 gene and, subsequently, the loss of its encoded protein, fragile X mental retardation 1 protein (FMRP) [54]. FMRP protein regulates the translation of hundreds of mRNAs in neuronal cells, most of which are involved in neuronal function and plasticity [54]. Among other functional consequences contributing to the pathogenesis of the disease, FMRP deficiency causes a dramatic increase in histone H3 and H4 acetylation in neuronal precursors and neurons coupled to a higher level of the histone acetylase enzyme EP300 and a lower level of HDAC1 [55]. A recent study showed that an acute and single intraperitoneal injection of SAHA in the mouse model of fragile X syndrome, *Fmr*1-KO, restores object location memory and passive avoidance memory and corrects repetitive behavior and social interaction deficits [51] (Figure 5). The authors have also analyzed the SAHA-induced transcriptome changes confirming its similarity to that induced by trifluoperazine, which was previously shown to correct pathological outcomes associated with FXS. However, although SAHA corrects several FXS-associated behavior symptoms, it does not normalize the elevated global protein synthesis observed in cultured Fmr1 KO hippocampal neurons; thus, the authors conclude that the elevated protein synthesis observed in the brain areas of *Fmr*1-KO mice is not linked to all behavioral abnormalities [51] (Table 2).

### 3.2. Autism Spectrum Disorders

Autism spectrum disorder (ASD) is a neurodevelopmental disorder affecting almost 1% of children, characterized by impairments in language and social interaction coupled with various degrees of intellectual disability (ID) [64]. Several studies have demonstrated a strong association between rare de novo and inherited copy number variants (CNV) with ASD. Therefore, it is logical to consider that a potential therapeutic intervention in ASD could be to identify a small molecule capable to rescue haploinsufficiency or overexpression of individual disease genes (Figure 5B). It has been reported that SAHA is able to correct gene expression of KDM5C, a disease gene involved in ID [14]. More recently, it has been shown that SAHA is capable to rescue gene expression in ASD patient-derived cortical neurons with 7q11.23 microduplication (7Dup) [32]. Furthermore, 7Dup is one of the best characterized ASD-causing CNVs encompassing 26–28 genes, some of which play a critical role in ASD pathogenesis [65]. One of them is general transcription factor II-I (*GTF2I*) which is the major mediator of the cognitive-behavioral alterations in 7Dup, responsible for a large part of transcriptional dysregulation. Of note, a symmetrically opposite hemi-deletion of the same region causes Williams–Beuren Syndrome (WBS), a multisystemic disease, including hypersociability and selectively spared verbal abilities, mild to moderate ID, and a severely compromised visual-spatial processing [66]. These two conditions caused by symmetrically opposite CNVs represent a unique genetic paradigm to dissect the dosage-vulnerable circuits affecting language processes and sociability. In this study, Testa and colleagues showed that SAHA restored the expression levels of key genes in 7Dup-induced neurons [32] (Figure 5B). In particular, by performing a high-throughput compound screening on cortical glutamatergic neurons differentiated from ASD-derived neurons, the authors analyzed the transcriptional modulation of 7Dup interval genes, establishing that SAHA decreased the abnormal expression level of GTF2I, at mRNA and protein levels [32]. In line with several studies demonstrating that HDACi activity can cause both up- and downregulation of gene expression, SAHA reduced *GTF2I* expression [32,67]. This dual activity of SAHA could depend on the involvement of enzymes or cofactors which in turn will act as activators or repressors of other downstream genes.

More recently, postnatal administration of low-dose SAHA in the ASD mouse model carrying a deletion of the *Ashl*1 gene, one of the high ASD risk genes identified in human patients [68], significantly ameliorated the core ASD-like behaviors of the Ash1L-deficient mice, such as sociability, repetitive behavior, and the defective cognitive memory [57]. Although the authors observed that SAHA administration did not alleviate anxiety-like or ataxia-like behaviors, they concluded that this epidrug could be considered a promising reagent for the pharmacological treatment of core ASD/ID behavioral and memory deficits caused by disruptive ASH1L mutations [57].

### 3.3. Epilepsy

In the last years, several studies have been performed to find new anticonvulsant drugs (AED) capable to stop and/or counteract the events transforming functional neuronal circuits into epileptic circuits (epileptogenesis) and consequently generating seizures (ictogenesis) [69,70,71]. Epilepsy is a neurological disorder that is characterized by abnormal and hypersynchronized electrical activity in neurons that in turn cause seizures and the associated cognitive and psychological abnormalities in patients [69]. Most of the genetic forms of epilepsy arise during the first months of life as a result of anomalies during neonatal development. Although many patients have seizure control using single or multiple medications or dietary therapies, one-third of patients continue to have uncontrolled pharmaco-resistant seizures. The applicability of HDACi for the treatment of epilepsy was introduced about 50 years ago with the introduction of valproic acid (VPA) as a potent first-line AED to treat child and adult patients with epilepsy, with either general or focal seizures, and in generalized convulsive status epilepticus [72]. This is an HDACi that broadly targets Class I and IIa HDACs through a broad spectrum of actions that may trigger several poorly tolerated side effects [73].

Recent studies have highlighted the promising use of SAHA as a potent anticonvulsant drug with limited off-target activity [53]. Since SAHA readily crosses the blood-brain barrier and is a molecule similar to VPA approved by therapeutic regulatory agencies worldwide for human internal use, it was tested as a new therapeutic drug for pediatric forms of epilepsy [53]. In epileptic zebrafish (kcna1-KO) and epileptic mice (Kcna1-null mice), ablated for the evolutionary conserved potassium channel gene *Kcna*1 that encodes the voltage-gated Kv1.1 potassium channel α-subunit, SAHA is capable to reduce daily seizure activity [53] (Figure 5C). In humans, mutations affecting the S1/S2 helices and pore domain of the Kv1.1 subunit have been detected in patients with epileptic encephalopathies (EE) characterized by early-onset seizures and epileptiform activity that progressively impairs brain function, leading to cognitive, behavioral, and language deficits [74]. In the study of Ibhazehiebo et al. [53], the authors showed that the perfusion of *kcna*1-KO zebrafish larvae with SAHA for 30 min completely abolished the high-frequency spikes and thus hyperexcitability. Furthermore, they treated spontaneously epileptic *Kcna*1-null mice with SAHA and observed, using video EEG, reduced seizure frequency by 60%, a phenomenon that started 24 h after the start of the treatment and took 4 days to reach significance, indicating that SAHA activity requires additional time to be effective [53]. In addition, by performing a metabolism-based phenotypic drug screening, the authors proved that SAHA rescues a peculiar metabolic profile associated with the “epileptic” brain in both animal models used. The anticonvulsant property of SAHA was also confirmed upon administration in the mouse model of tuberous sclerosis complex (TSC), an autosomal dominant genetic disorder with seizures, autism, and cognitive deficits [75]. It has been estimated that approximately 85% of TSC patients exhibit seizures within the first year of life and many of them exhibit intractable epilepsy [75]. By administration in TSC+/− mice presenting defects in long-term memory formation and potentiation and a greater susceptibility to developing seizure, SAHA is capable to restore hippocampal plasticity and normalize seizure threshold phenotype resembling a WT response [63]. Indeed, starting from the unbalance between histone deacetylase and acetylase activity detected in TSC2+/− brain areas, the authors proved that the pharmacological inhibition of HDAC activity via SAHA restores the H3 acetylation levels (H3K9Ac and H3K27Ac), ameliorating the aberrant synaptic plasticity and the seizure threshold [73]. As for the adverse effects of SAHA as an antiepileptic agent, no data on patients have been reported to date. Related to VPA, the most frequent adverse effects reported are nausea, vomiting, fatigue, fever, headache, and anorexia [72]. Because the benefits of AED are more major than the adverse effects, VPA represents a drug of choice in children and adults with epilepsy, with general or focal seizures [72]. Remarkably, VPA is not recommended as an anticonvulsant drug in pregnant women with epilepsy because crossing the placenta has teratogenic activity [72]. Thus, given the similarity with VPA, it is fundament to verify in further studies whether SAHA may also cause congenital abnormalities.

### 3.4. Depression

It is a psychiatric disorder affecting millions of people worldwide, mainly characterized by depressed mood, diminished interest in ordinary activities and life aspects, impaired cognitive function, and sleep and appetite perturbations [76]. Even though the pathophysiology of depression is still not completely understood, it is certainly caused by multiple factors, including genetics and epigenetic factors [59]. Several preclinical studies have been reported to demonstrate the antidepressant-like efficacy of SAHA acting as a molecule capable to remodel the chromatin structure and to counteract transcription alterations affecting neurotrophin genes, such as *Bdnf* and *Gdnf* [52,59] (Figure 5B). Antidepressant activity of SAHA has also been found to ameliorate depressive symptoms during alcohol withdrawal [60] and more recently in a murine model for maternal separation (MS) effects [61]. The exact mechanisms through which SAHA exerts its antidepressant activity are poorly known. A number of studies showed that this epidrug has a pivotal role in modulating oxidative stress pathway activity in the prefrontal cortex and hippocampus, two regions mainly involved in depressive behavior [77,78]. Remarkably, in acute and chronic SAHA treatments, a reduction of lipid peroxidation with an increase in p-AMPK and HDAC3 and of the synaptic plasticity proteins GluN2A and GluN2B have been observed [78].

## 4. SAHA in Other Neurological Disorders

The discovery of the molecular basis of different neuronal signaling pathways underpinning neuropathology reveals the active participation of acetylating and deacetylating enzymes at the onset of the disease and during its progression.

In this context, accumulating evidence highlights the beneficial role of SAHA in different neurodegenerative pathologies (Table 2) [6]. Recent studies have proved the combined use of SAHA with other chemical compounds improves its therapeutic efficacy [56]. In a mouse model of Alzheimer’s disease (AD), it has been demonstrated that SAHA is able to enhance the therapeutical activity of rosiglitazone, an agonist of the peroxisome proliferator-activated receptor (PPAR) family, already proposed as a promising drug in AD treatment [56]. The authors showed that the administration of SAHA in combination with rosiglitazone more efficiently attenuates different behavioral alterations, compared with single rosiglitazone treatment [56].

In addition, they demonstrated that the combined formulation of SAHA with rosiglitazone significantly reduced both the expression levels of the neuroinflammatory markers IL6 and TNF-a and the oxidative stress markers MDA and nitrite and restored GSH and SOD levels [56]. In line with the neuroprotective activity of SAHA, several studies carried out in different transgenic murine models for neurodegenerative diseases confirm this capacity. Of particular interest are the results obtained in R6/2 mice, a model for Huntington’s disease (HD) with CAG repeat expansions in Huntingtin (HTT) gene, in which SAHA treatment reinstates transcriptional repression—*via* global histone acetylation [15]—and reduces HTT-expanded aggregates with the restoration of BDNF levels in the cortex [58]. Interestingly, a recent study describes the beneficial effects of SAHA in a mouse model of retinitis pigmentosa (RP), which is a progressive neurodegenerative disease of the retina. Indeed, in an RP-rd1 mouse model ablated in the rhodopsin gene whose mutations cause a recessive form of retinitis pigmentosa (RP), SAHA treatments increased the photoreceptor cell survival and improved mitochondrial respiration in the retina [62]. In a more recent study, SAHA treatments carried out in patient-derived myoblasts with myotonic dystrophy type 1 (DM1) and in a murine DM1 model reduced RNA foci formation and thus restored aberrant splicing [23]. This capacity of SAHA to counteract the defects caused by aberrant alternative splicing isoforms has also been detected in the mouse model for the autosomal dominant deafness 27 (DFNA27; OMIM 612431), caused by mutations in the *REST* gene [25]. In humans, a REST variant, which inhibits the splicing event of *REST* exon 4, was found in patients of a family (LMG2) with an inherited dominant form of postlingual progressive hearing loss [79]. Specifically, the genome-wide scan in a 3-generation family linked autosomal dominant hearing loss to the *REST*-containing DFNA27 locus on chromosome 4q12-q13.1 [79]. Subsequently, by sequencing of conserved intronic regions of the candidate gene REST, Nakano et al. [25] identified a heterozygotic intronic variant (NC_000004.12:g.56927594CG) that segregated fully with deafness in the family and was causative of the prevention of exon 4 skipping, which was necessary for the depression of many neuronal genes [80]. The authors proved the ability of SAHA to increase the expression of REST target genes that were downregulated in *Rest*^+/ΔEx4^ mice and to correct the stereocilia in the inner and outer hair cells of organs of Corti cultures. More interestingly, the systematic administration of SAHA in REST exon 4-deficient newborn mice—when the BBB is still not fully developed—was able to strongly improve the formation of stereocilia in the outer hair cells of the Corti organ [25]. Recently, in human glioblastoma cell lines, a peculiar antiangiogenic property of SAHA has been observed, capable to inhibit vasculogenic mimicry by interfering with the differentiation status of GBM cells [81], perhaps by altering the expression of neuronal markers that in turn trigger the epithelial-mesenchymal transition, as also observed in GBM cell lines at low concentrations of the HDACi trichostatin A (TSA) [82].

## 5. Conclusions

The combination of neuroprotective and neuritogenic effects observed in SAHA strengthens its potential for drug repurposing in the treatment of cognition impairment and epilepsy, overcoming contradictory observations reported in the previous literature about its neuroprotective or neurotoxic effects on the CNS that may depend on target cells and tissues, as well as the dosage used [38,39]. Indeed, in the last decades, we have had a clear advancement in our understanding of the beneficial activity of SAHA as a neuroplasticity driver in models of various neurological diseases, acting through a variety of molecular processes, for example, by increasing neurotrophin levels (e.g., *Bdnf*) or counteracting oxidative stress [60,61,77,78].

The diversity of SAHA-sensitive pathways is not a surprise because this HDACi has two wide effects: (i) it induces hyperacetylation of histones H3 and H4 modifies global chromatin compaction and transcriptional gene expression; and (ii) it induces hyperacetylation of nonhistone proteins involved in different cellular functions, including neurogenesis and synaptogenesis. However, the effects of SAHA on the course of different neurological diseases are only partly understood, and further studies in complementary preclinical models and in patient-derived cortical brain organoids that recapitulate *in vivo* features of NDD phenotypes may clarify the promising therapeutic applications.

Given the potential of SAHA as a particularly promising repurposed compound for NDDs, a better comprehension of the mechanism(s) of action may represent the road ahead for personalized medicine and insights into potential pharmacological approaches to modulating neuronal functions during the early stages of the disease.

## Figures and Tables

**Figure 1 biomolecules-13-01301-f001:**
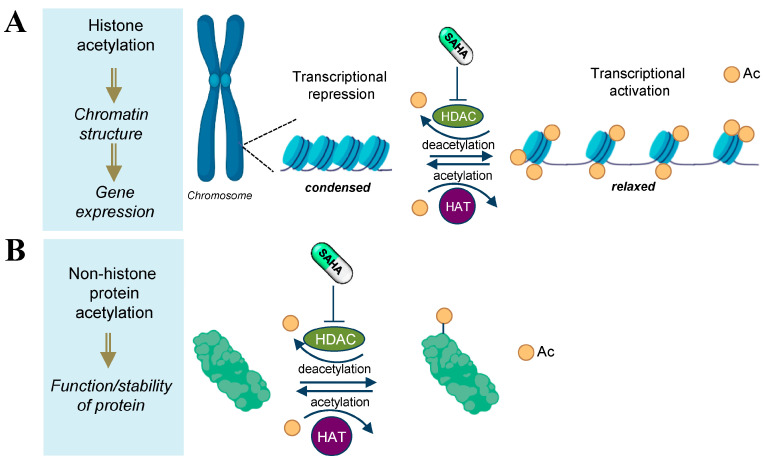
Schematic representation of the molecular pathways involved in HDAC inhibition by SAHA. The acetylation status of histones is regulated by the opposing action of histone acetyl transferases (HATs) and histone deacetyl transferases (HDACs). SAHA inhibits HDACs and induces hyperacetylation of both histone (**A**) and nonhistone proteins (**B**) [3].

**Figure 2 biomolecules-13-01301-f002:**
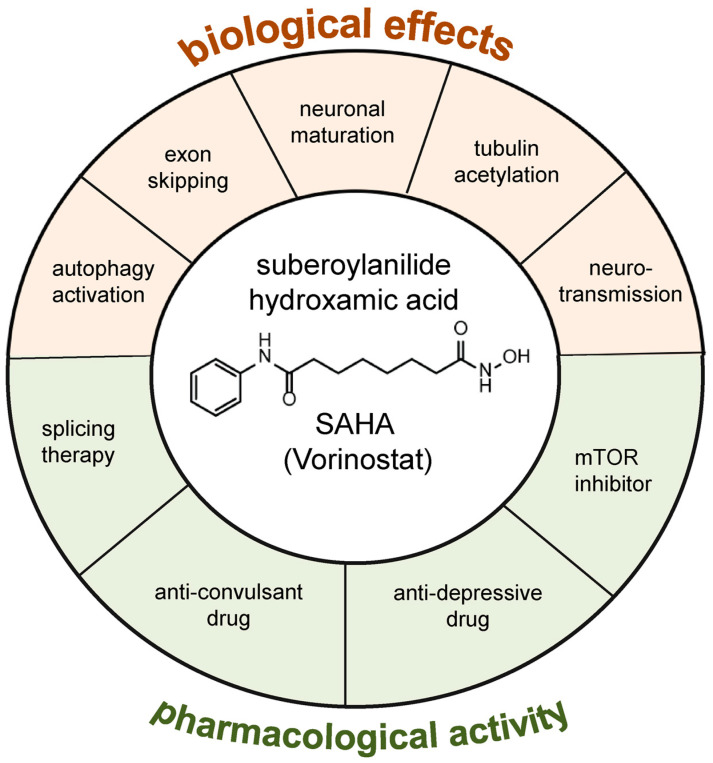
Overview of the molecular and biological effects of SAHA as a potential plasticity driver molecule. In red, SAHA-sensitive processes involved in neuronal functions; in green, the pharmacological activity of SAHA in neurological disorders.

**Figure 3 biomolecules-13-01301-f003:**
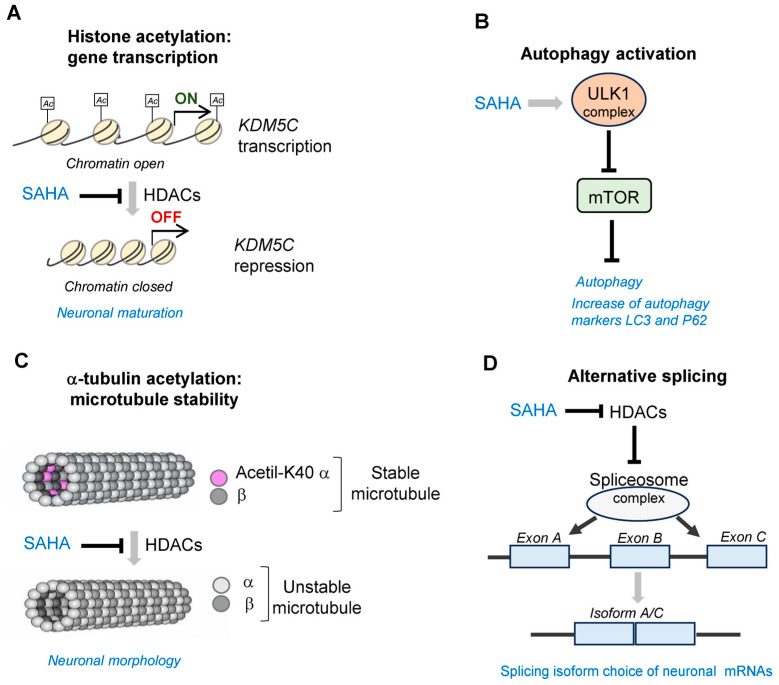
The multiple activities of SAHA as a neuroactive compound. (**A**). Transcription of genes involved in neuronal maturation, e.g., KDM5C: SAHA inhibits HDAC activity favoring the openness of chromatin and thus gene transcription [14]; Ac = acetylation. (**B**). Activation of autophagy regulating the protein levels of the synaptic components and, thus, the synaptic plasticity: SAHA increases the activity of ULK1 complex, which is the most upstream component in the core autophagy pathway and suppresses the mammalian target of rapamycin (mTOR), which in turn inhibits autophagy induction. SAHA also induces the transcription of the autophagy markers LC3 and P62 [19,21,27]. (**C**). Stabilization of the structure of microtubules, essential regulators of neuronal cytoarchitecture: SAHA inhibits the HDAC activity keeping the levels of acetylation of alpha-tubulin monomers, a posttranslational modification that stabilizes the microtubule structure [11,22]; (**D**). Formation of the spliceosome complex required to regulate splicing choices occurring during brain development and in mature neurons: SAHA inhibits HDAC activity and favors the formation of spliceosome complex with the generation of the isoform A/C [24].

**Figure 4 biomolecules-13-01301-f004:**
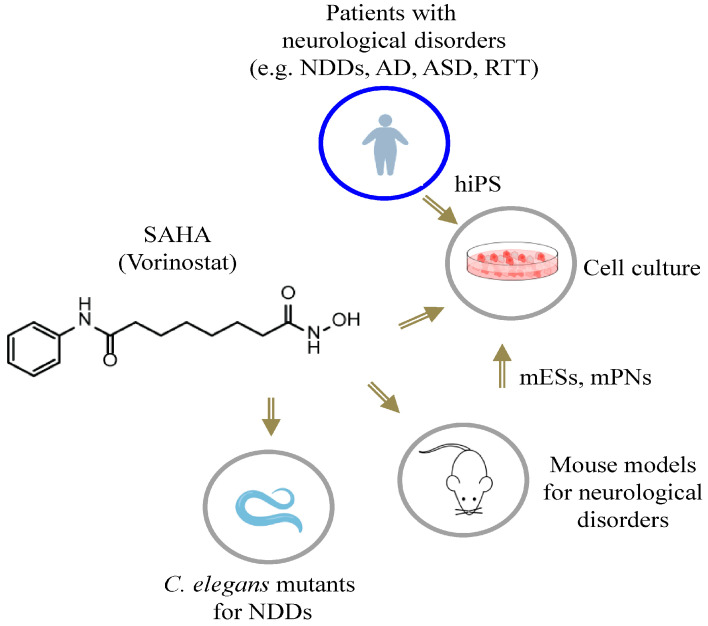
SAHA treatments in cellular and animal models for neurological disorders. Schematic representation of the tools used. NDDs, neurodevelopmental disorders; AD, Alzheimer’s disease; ASD, autistic spectrum disorders; RTT, Rett syndrome; hiPSs, human-induced pluripotent stem cells; mESs, murine embryonic stem cells; mPNs, murine primary neurons.

**Figure 5 biomolecules-13-01301-f005:**
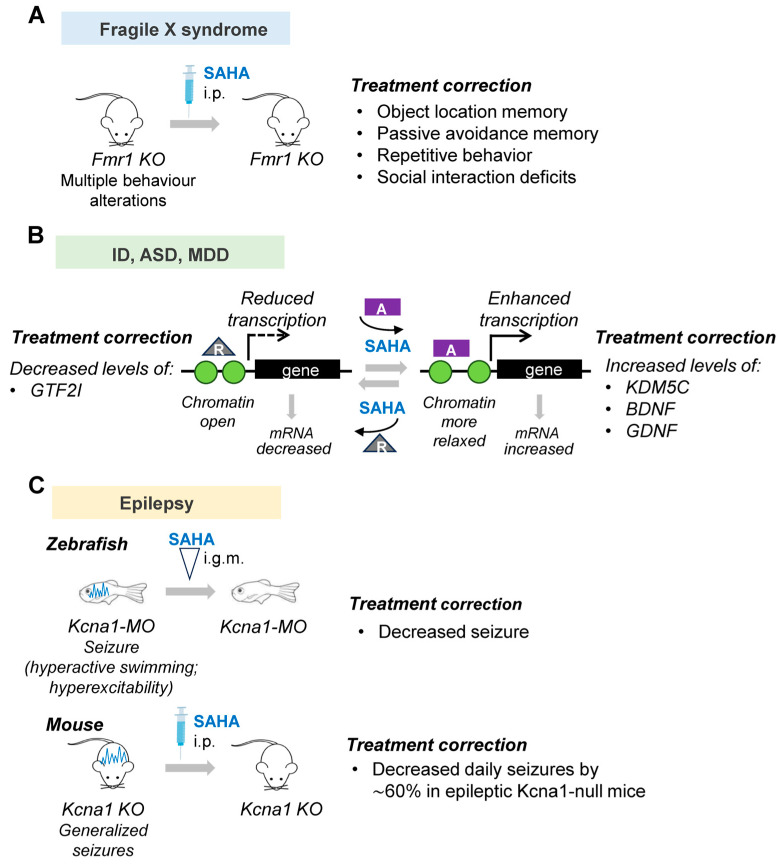
Functions of SAHA in models of neurodevelopmental and psychiatric diseases: (**A**). Correction of autism-associated alterations, including repetitive behavior and social interaction deficits in fragile X syndrome (FXS) Fmr1 KO mice [51]. (**B**). Rescue of transcriptional defects affecting neuronal genes involved in ID (e.g., KDM5C, FMR1), ASD (e.g., GTF2I), and MDD (e.g., BDNF, GDNF) [14,32,51,52]; (**C**). Seizure frequency reduction in zebrafish and mouse *Kcna*1 null-mutants [53]. ID = intellectual disability; ASD = autism spectrum disorder; MDD = major depressive disorder; MO = morpholine; i.g.m = injection by a glass microelectrode in the tectum opticum; i.p. = intraperitoneal injection.

**Table 1 biomolecules-13-01301-t001:** Molecular functions of SAHA as a neuroactive compound.

Mechanism	Molecular Pathway/Mechanism of Action	Ref.
Induction of neuronal differentiation	Restoring of KDM5C-H3K4me3 signaling; induction of *Bdnf* transcription	[14]
Regulation of autophagy	Activation of the ULK1 complex; suppression of mTOR; increase in the autophagy markers LC3 and P62	[19,21,27]
Regulation of axonal transport of BDNF	Increase in α-tubulin acetylation, enhancing kinesin-1 motility along MTs	[22]
Increase in neurite outgrowth	ERK phosphorylation; increase in α-tubulin and H3 and H4 histones acetylation	[11]
Memory formation	Induction of *Bdnf* transcription	[28,29]
Neuronal morphogenesis and synaptic plasticity	Increase in the expression of plasticity-related genes	[30,31,32]
Inhibition of axonal damage	HSP70 acetylation	[33]

**Table 2 biomolecules-13-01301-t002:** SAHA in preclinical models of neurodevelopmental, psychiatric, and neurodegenerative diseases.

Human Disease	Disease Models	Molecular Pathway/Mechanism of Action	Ref.
Alzheimer’s disease (AD) [MIM: 104300]	AD mice	improvement of the oxidative stress response; increase in neurotrophic factor levels; improvement of cognitive functions (in combination with Rosiglitazone)	[56]
Autism spectrum disorder (ASD) [MIM: 209850]	7Dup iPSC-derived cortical neurons	correction of the aberrant transcriptional level of GTF2I	[32]
	Ash1L-Nes-cKO mice	amelioration of AS-like behaviors and ID phenotype	[57]
Deafness autosomal dominant 27 (DFNA27) [MIM: 612431]	Rest-exon 4 deficient mice	increase in REST-target gene expression; rescue of inner and outer hair cells stereocilia formation; partial rescue of hearing defects	[25]
Episodic ataxia/myokymia syndrome [MIM: 176270]	Kcna1-null zebrafish Kcna1-KO mice	seizure frequency reduction	[53]
Fragile X syndrome (FXS) [MIM: 300604]	FMR1 KO mice	amelioration of memory, repetitive behavior, and social interaction deficits	[51]
Huntington’s disease (HD) [MIM: 143100]	HTT- R6/2 mice	improvement of motor impairments; decrease in insoluble aggregates in the cortex and brain stem; restoration of *Bdnf* transcript levels in the cortex	[15,58]
Major depressive disorder (MDD) [MIM: 608516]	chronic-stress mice model	correction of defective neurotrophin levels; rescue of depressive-like behavior	[52,59]
	alcohol withdrawal stress rat model	rescue of H3K9 acetylation in hippocampus; rescue of depressive-like behavior	[60]
	maternal separation stress mice model	rescue of depressive-like behavior and counteraction of neuroinflammation	[61]
Myotonic Dystrophy Type 1 (DM1) [MIM: 160900]	DM1 patient myoblast DM1- HSASR mice	inhibition of RNA foci formation and release of MBNL1 splicing factor; improvement of aberrant mRNA splicing	[23]
Retinitis Pigmentosa (RP) [MIM: 268000]	RP-rd1 mice	improvement of mitochondrial respiration in photoreceptor cells; increase in photoreceptor cell survival	[62]
Tuberous sclerosis (TSC) [MIM: 191100]	TSC2+/− mice	amelioration of synaptic plasticity and reduction of seizure threshold	[63]

## Data Availability

Not applicable.

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
