# Peer review of "Suberoylanilide Hydroxamic Acid (SAHA) Is a Driver Molecule of Neuroplasticity: Implication for Neurological Diseases"

_biomolecules, 2023, doi:10.3390/biom13091301_

Round 1

Reviewer 1 Report

Verrillo et al. reviewed suberoylanilide hydroxamic acid (SAHA) as a useful agent to improve memory, learning, and behavior and to correct deficits in neuronal function. However, there are still several points that the authors need to correct before publication in "Biomolecules."

1) Histone deacetylases (HDACs) are classified as Class I, Class IIa, Class IIb, and Class IV, with at least 11 subtypes. As stated by the authors, SAHA is a pan-HDAC inhibitor that inhibits both Class I and Class II HDACs. Therefore, the advantages and, if possible, disadvantages of pan-HDAC inhibitors, SAHA, compared to class-specific and/or subtype-specific HDAC inhibitors in terms of neuroplasticity should be discussed.

Indeed, a number of review articles on HDAC inhibitors for neurological diseases have been published in the last few years. Since this review deal with SAHA among the various HDAC inhibitors, it should also discuss the advantages and, if possible, disadvantages of SAHA over other HDAC inhibitors.

2) HDAC inhibitors are known to act broadly not only transcriptionally but also post-translationally, however the authors only described the post-translational effect, the non-histone protein acetylation, of SAHA on alpha-tubulin. It has been reported that many proteins are regulated by the acetylation/deacetylation. The authors should discuss the effects on the function of various non-histone proteins acetylation by SAHA treatment from the view of neuroplasticity.

3) Since, not only SAHA but also other several HDAC inhibitors have been already clinically used as anti-cancer drugs, and valproic acid, which was reported to inhibits HDAC, has been used as a classic anti-epilepsy drug for a long time, the authors should discuss the both effects and adverse effects on neuroplasticity of actually used drugs with referencing to not only in vitro and in vivo data but also clinical data of them.

Author Response

Comments and Suggestions for Authors

Verrillo et al. reviewed suberoylanilide hydroxamic acid (SAHA) as a useful agent to improve memory, learning, and behavior and to correct deficits in neuronal function. However, there are still several points that the authors need to correct before publication in "Biomolecules."

Q1) Histone deacetylases (HDACs) are classified as Class I, Class IIa, Class IIb, and Class IV, with at least 11 subtypes. As stated by the authors, SAHA is a pan-HDAC inhibitor that inhibits both Class I and Class II HDACs. Therefore, the advantages and, if possible, disadvantages of pan-HDAC inhibitors, SAHA, compared to class-specific and/or subtype-specific HDAC inhibitors in terms of neuroplasticity should be discussed.

Indeed, a number of review articles on HDAC inhibitors for neurological diseases have been published in the last few years. Since this review deal with SAHA among the various HDAC inhibitors, it should also discuss the advantages and, if possible, disadvantages of SAHA over other HDAC inhibitors.

R1) We thank the reviewer for this comment. The definition of advantages and disadvantages of pan-HDACIs require a very accurate discussion in a review that include specifically pharmacology studies, pre-clinical and clinical studies, clinical trials published to date. Part of this aspect was discussed in other reviews published in the last years, giving a very exhaustive description (Goey et al. 2016, Suresh et al. 2017; Shukla et al. 2020; Bondarev et al. 2021; Athira et al. 2021; Shetty et al. 2021). Among them, we found very interesting the review by Athira et al. 2021 in which the authors have discussed about the safety and tolerability of SAHA declaring that it is fundamental the characterization of the changes induced by different HDACs in order to identify the proper HDAC treatment for a specific patient. In addition, they highlight the reversible activity of SAHA supporting the concept that the optimization of the therapeutic strategies with respect to dose, dosage regimen, and formulations of SAHA could propel its clinical prospects. The main point of our review is not discussed again this point, but to highlight the neuroplasticity property of SAHA that could be tested in animal models of neurological disorders that has not been tested to date, giving an overview of cellular and animal models tested, functional processes analysed, molecular pathways sensitive to SAHA treatments.

Nevertheless, based on the Reviewer’s comments and our considerations, we have added the following sentence in the revised MS Verrillo et al. R1:

Section Introduction, Line 55, page 2:

“SAHA is a pan-HDAC inhibitor that inhibits both Class I and Class II HDACs. It belongs to a wide group of HDACI that vary in structures, target selectivity and biological activities. In the recent years, their HDAC inhibitor activity has been exploited as anti-proliferative and pro-apotopic agents in the treatments of cancer and as neuroprotective and neurotrophic molecules in non-malignant conditions of the nervous system (Goey et al. 2016; Suresh et al. 2017; Shukla et al. 2020; Bondarev et al. 2021; Athira et al. 2021; Shanmugam et al. 2022). This dual activity mirrors the lack of molecular specificity and indicates that the pharmacological effects of SAHA and other HDACIs are based on a wide range of mechanisms of action -a feature common to several drugs- which identification require further studies. Of course, given the global effects of SAHA and other HDACIs, it is fundamental to distinguish benefit and detrimental effects, both in vitro and in vivo. Several studies highlight that, similarly to other HDACIs, the therapeutic outcome of SAHA -as effect, efficacy, effectiveness and benefit- may vary depending to the dosage used and the biological context analysed (e.g. cell type, tissue specificity, developmental window, etc) (Athira et al. 2021; Poeta et al. 2019; Hockly et al. 2003; Benito et al 2015; Gottesfeld et al. 2009; Elvir et al. 2019; Lunke et al. 2021). Although future research may elucidate the downstream effects of SAHA, there is a clear trend towards its applicability with new drug repurposing opportunities because it has: i. a good blood brain barrier permeability; ii. does not lead to massive transcriptional changes; and iii. compared to others HDACi, it is active at nanomolar concentrations in a variety of cellular and animal models (Athira et al. 2021; Hockly et al. 2003; Benito et al 2015; Gottesfeld et al. 2009; Elvir et al. 2019; Lunke et al. 2021). Future multi-omic studies in in vivo models coupled to fast transcriptome and proteome screenings may help to figure out the pharmacology response of SAHA to distinguish at molecular levels the advantages and disadvantages of SAHA, also compared to other HDAC inhibitors, allowing the adoption of a personalized applicability of HDAC activity inhibition. Although pharmacogenomic studies for HDAC inhibitors are still scarce, genotype-directed dosing could improve pharmacotherapy of SAHA reducing toxicity risk or suboptimal dosage. In view of more in-depth studies on the pharmacology response of the SAHA, the main point of this review is to highlight the neuroplasticity property of SAHA that could be tested in animal models of neurological disorders that has not been tested to date, giving an overview of cellular and animal models tested, functional processes analysed, molecular pathways sensitive to this epi-treatment. In particular, ….”

New references included:

Bondarev AD, Attwood MM, Jonsson J, Chubarev VN, Tarasov VV, Schiöth HB. Recent developments of HDAC inhibitors: Emerging indications and novel molecules. Br J Clin Pharmacol. 2021 Dec;87(12):4577-4597.

Benito, E., Urbanke, H., Ramachandran, B., Barth, J., Halder, R., Awasthi, A., Jain, G., Capece, V., Burkhardt, S., Navarro-Sala, M., Nagarajan, S., Schütz, A. L., Johnsen, S. A., Bonn, S., Lührmann, R., Dean, C., & Fischer, A. HDAC inhibitor-dependent transcriptome and memory reinstatement in cognitive decline models. The Journal of clinical investigation, 2015, 125, 3572–3584

Goey AK, Sissung TM, Peer CJ, Figg WD. Pharmacogenomics and histone deacetylase inhibitors. Pharmacogenomics. 2016 Nov;17(16):1807-1815. doi: 10.2217/pgs-2016-0113.

Shanmugam et al. HDAC inhibitors: Targets for tumor therapy, immune modulation and lung diseases,Translational Oncology,Volume 16,2022.

Shetty MG, Pai P, Deaver RE, Satyamoorthy K, Babitha KS. Histone deacetylase 2 selective inhibitors: A versatile therapeutic strategy as next generation drug target in cancer therapy. Pharmacol Res. 2021 Aug;170:105695.

Suresh PS, Devaraj VC, Srinivas NR, Mullangi R. Review of bioanalytical assays for the quantitation of various HDAC inhibitors such as vorinostat, belinostat, panobinostat, romidepsin and chidamine. Biomed Chromatogr. 2017 Jan;31(1). doi: 10.1002/bmc.3807. Epub 2016 Sep 13. PMID: 27511598

Q2) HDAC inhibitors are known to act broadly not only transcriptionally but also post-translationally, however the authors only described the post-translational effect, the non-histone protein acetylation, of SAHA on alpha-tubulin.

It has been reported that many proteins are regulated by the acetylation/deacetylation.

The authors should discuss the effects on the function of various non-histone proteins acetylation by SAHA treatment from the view of neuroplasticity.

R2) We thank for this comment.

About the effects of SAHA on the acetylation of non-histone proteins -as p53, Bcl6, Stat3 and others- it has been reported this activity in cancer diseases (Singh et al. 2010; Yang et al. 2022), but how the effects on these proteins may contribute to modulate/modify proteins involved in neuronal plasticity, in health and disease conditions, is still unknown.

On the contrary, it is well described the SAHA effects on alpha-tubulin. However, since recently it has been described the role of SAHA to prevent axon degeneration in a rat model of subarachnoid haemorrhage acting as heat shock protein (HSP) co-inducer via the acetylation of HSP70, we have added a new comment in the revised MS Verrillo et al. R1:

Section 2.3 “Microtubule organization”, line 206, page 6:

“Related to the acetylation/deacetylation effect of SAHA, Luo et al [2022] described a new mechanism of action involved in preventing axonal damage and neurological dysfunction induced in a rat model of subarachnoid hemorrhage (SAH). This SA-HA-induced response is mediated by heat shock protein (HSP) co-inducer activity via the acetylation of HSP70, which in turn induces the degradation of inclusion bodies formed by TDP-43 aggregates [Luo et al 2022].”

We also added in Table 1 a new row related to this new comment and the new reference

New reference included:

Luo, K., Wang, Z., Zhuang, K. et al. Suberoylanilide hydroxamic acid suppresses axonal damage and neurological dysfunction after subarachnoid hemorrhage via the HDAC1/HSP70/TDP-43 axis. Exp Mol Med 54, 1423–1433 (2022).

Q3) Since, not only SAHA but also other several HDAC inhibitors have been already clinically used as anti-cancer drugs, and valproic acid, which was reported to inhibits HDAC, has been used as a classic anti-epilepsy drug for a long time, the authors should discuss the both effects and adverse effects on neuroplasticity of actually used drugs with referencing to not only in vitro and in vivo data but also clinical data of them.

R3) We thank the reviewer for this comment.

After more than a century from its discovery, valproic acid or VPA still represents one of the most efficient anti-epileptic drugs. As extensively reported, the most frequent adverse effects caused by VPA are nausea, vomiting, fatigue, fever, headache and anorexia. However, because the benefits as AED are major than the adverse effects, VPA is represents a drug of choice in child and adult with epilepsy, with either general or focal seizures. Remarkably, VPA is not-recommended as anti-convulsant drug in pregnant women with epilepsy because it crosses the placenta and has teratogenic activity.

Given these considerations, we have added this comment:

Section 3.3 Epilepsy, lane 382, page 11:

“About the adverse effects of SAHA as anti-epileptic agent, to date no data on patients have been reported. Related to VPA, the most frequent adverse effects reported are nausea, vomiting, fatigue, fever, headache and anorexia [Romoli et al. 2019]. Because the benefits as AED are major than the adverse effects, VPA represents a drug of choice in child and adult with epilepsy, with general or focal seizures [Romoli et al. 2019]. Remarkably, VPA is not-recommended as anti-convulsant drug in pregnant women with epilepsy because it crosses the placenta and has teratogenic activity [Romoli et al. 2019]. Thus, given the similarity with VPA, it is fundament to verify whether SAHA may also cause congenital abnormalities”.

Reviewer 2 Report

In this review, the authors summarize the various mechanisms by which the HDAC inhibitor SAHA improves memory, learning, behavior, and corrects faulty neuronal functioning in different brain disorders. Additionally, the authors conclude that SAHA promotes positive regulation of neuronal gene expression, microtubule dynamics, neurite outgrowth, spine density, and enhances synaptic transmission and potentiation. The manuscript is well-organized, and I recommend accepting it for publication with minor revisions.

To provide a comprehensive understanding of the mechanism of SAHA in neuronal maturation and plasticity, activation of autophagy, microtubule organization, and regulation of alternative splicing switch, detailed explanations with illustrations are necessary. Additionally, illustrative images are needed to fully explore the function of SAHA in neurodevelopmental and psychiatric diseases.

Author Response

Comments and Suggestions for Authors

In this review, the authors summarize the various mechanisms by which the HDAC inhibitor SAHA improves memory, learning, behaviour, and corrects faulty neuronal functioning in different brain disorders. Additionally, the authors conclude that SAHA promotes positive regulation of neuronal gene expression, microtubule dynamics, neurite outgrowth, spine density, and enhances synaptic transmission and potentiation. The manuscript is well-organized, and I recommend accepting it for publication with minor revisions.

Q1) To provide a comprehensive understanding of the mechanism of SAHA in neuronal maturation and plasticity, activation of autophagy, microtubule organization, and regulation of alternative splicing switch, detailed explanations with illustrations are necessary. Additionally, illustrative images are needed to fully explore the function of SAHA in neurodevelopmental and psychiatric diseases.

R1). First of all, we thank the Reviewer for her/his consideration. Following her/his suggestions we have added two new figures in Verrillo et al. R1:

Figure 3. The multiple activities of SAHA as neuroactive compound

Figure 5. Functions of SAHA in models of neurodevelopmental and psychiatric diseases

In both new figures we have included illustrations hoping to ameliorate the message of our review. The figure 3 of the original MS has become figure 4 in the revised version of MS.

We have also added the following sentences:

Section 3.2 Autism spectrum disorders, lane 304, page 9:

“….or overexpression of individual disease genes (Fig. 5B). It has been reported that SAHA is able to correct gene expression of KDM5C, a disease gene involved in ID [14]. More recently, it has been shown that SAHA is capable to rescue gene expression in ASD…”

Section 3.2 Autism spectrum disorders, lane 323, page 10:

“In line with several studies demonstrating that HDACi activity can cause both up- and downregulation of gene expression, SAHA reduces GTF2I expression [37; Thomas et al. 2008]. This dual activity of SAHA could depend on the involvement of enzymes or cofactors which in turn will act as activators or repressors of other downstream genes”.

New reference included:

Thomas EA, Coppola G, Desplats PA, Tang B, Soragni E, Burnett R, et al. The HDAC inhibitor 4b ameliorates the disease phenotype and transcriptional abnormalities in Huntington’s disease transgenic mice. Proc Natl Acad Sci. 2008;105:15564–9.

Reviewer 3 Report

The authors have addressed the issue of the role exerted by SAHA, an HDAC pan inhibitor, on neuroplasticity. Indeed, neuroplasticity is capable to confer resistance and/or to ameliorate not only neurodegenerative diseases but also neurodevelopment disorders, epilepsy and depression. Thus this issue is of general interest to basic and clinical researchers working on different fields in neurobiology, neuropathology and clinical neurology.

On the whole, this review is well-written and organized and easy to follow. It provides an in-depth information of the state-of art of the role of SAHA in neuroplasticity and neuropathology. 

However, some minor changes and the development of some points would improve the quality of this ms and would help the readers to better understand some concept that have been given without many details.

1. First of all, the ms deals with SAHA and neuroplasticity in the context of neurological diseases. Indeed, neuroplasticity has not been covered as such but as a mechanism targeted by SAHA and this in turn is able to interfere with the pathogenesis of some neurological diseases. For this reason, the title would better reflect the content of the ms if it were changed, for example,  as follow: "SAHA is a driver molecule of neuroplasticity: implication for neurological diseases".

2.  The authors mentioned the role of ERK phosphorylation as pro-neural molecule and in mediating  neuroplasticity upon SAHA administration. This is extremely interesting but it deserves to be better developed since in some instances ERK activation may lead to detrimental effects  and my contribute to neurodegeneration, whereas in some other cases it fosters survival, differentiation or is inhibited following detrimental mutations such as in HD.  There are many examples regarding this dual role of ERK, for instance ERK activation plays a detrimental role in glutamate neurotoxicity (Stanciu et al JBC  2000), in PD (Gómez-Santos C,  et al MPP+ increases alpha-synuclein expression and ERK/MAP-kinase phosphorylation in human neuroblastoma SH-SY5Y cells. Brain Res. 2002)  in  AD ( Nizzari et al JAD 2012; Chun, Y.S., et al Mol Psychiatry 2022). On the other hands, the positive role of  ERK in neuron survival, in memory formation as well as its inhibition in neurodegenerative diseases is well documented (Medina et al 2018; Bodai et al 2012; Melone et al 2013; Fusco FR et al 2012). It should be pointed out that a dual role of erk depends on the kinetics of activation and on its cellular compartmentalization ( colucci-d'amato  2003). Thus the authors would provide a better signaling context for SAHA effects, highlighting that, according to the literature, it is likely that  SAHA  not only activates ERK but does it in a pro-survival mode.

3. In the section, "SAHA in other neurological diseases" the authors overlooked an important effect of SAHA , that is the ability to inhibit vasculogenic mimicry in GBM , likely by altering neural transdifferention into endothelial-like cells.This  has to do with plasticity thus it is pertinent to the review. It is worth to add this part in the text as well as in Fig.2 (biological effects: vasculogenic mimicry) and in Table 2 (Human disease : GBM)

4. In the Introduction: from a lane 28  (Neuroplasticity...)to  lane 35 (synaptic network..). A reference should be provided 

5. In Concluding Remarks : lanes 384-386: "The combination ....overcoming contradictory information reported in the previous literature ". examples should be given

  Minor points:

1. legend to fig. 1 : line 46 the article is missing

2. pag 2 line 56 the abbreviation ID has been cited for the first time but there is no explanation

3. pag3 line 68 ...."in red": I should be capital letter

4. in authors contributions the name of Alberto de Bellis, one of the coauthor is absent.

To conclude, I think that this review deserves publication and would be interesting for a broad audience. Thus, the suggested points would help readers with different backgrounds to better understand the role of SAHA in neuroplasticity and well as in brain diseases.

the quality of english is good

Author Response

Comments and Suggestions for Authors

The authors have addressed the issue of the role exerted by SAHA, an HDAC pan inhibitor, on neuroplasticity. Indeed, neuroplasticity is capable to confer resistance and/or to ameliorate not only neurodegenerative diseases but also neurodevelopment disorders, epilepsy and depression. Thus, this issue is of general interest to basic and clinical researchers working on different fields in neurobiology, neuropathology and clinical neurology.

On the whole, this review is well-written and organized and easy to follow. It provides an in-depth information of the state-of art of the role of SAHA in neuroplasticity and neuropathology. 

However, some minor changes and the development of some points would improve the quality of this ms and would help the readers to better understand some concept that have been given without many details.

Q1) First of all, the ms deals with SAHA and neuroplasticity in the context of neurological diseases. Indeed, neuroplasticity has not been covered as such but as a mechanism targeted by SAHA and this in turn is able to interfere with the pathogenesis of some neurological diseases. For this reason, the title would better reflect the content of the ms if it were changed, for example, as follow: "SAHA is a driver molecule of neuroplasticity: implication for neurological diseases".

R1) Thank you for the suggestion. We appreciate it and change the title “Suberoylanilide hydroxamic acid (SAHA) is a driver molecule of neuroplasticity” into “Suberoylanilide hydroxamic acid (SAHA) is a driver molecule of neuroplasticity: implication for neurological diseases”.

Q2) The authors mentioned the role of ERK phosphorylation as pro-neural molecule and in mediating neuroplasticity upon SAHA administration. This is extremely interesting, but it deserves to be better developed since in some instances ERK activation may lead to detrimental effects and may contribute to neurodegeneration, whereas in some other cases it fosters survival, differentiation or is inhibited following detrimental mutations such as in HD.  There are many examples regarding this dual role of ERK, for instance ERK activation plays a detrimental role in glutamate neurotoxicity (Stanciu et al JBC  2000), in PD (Gómez-Santos C et al MPP+ increases alpha-synuclein expression and ERK/MAP-kinase phosphorylation in human neuroblastoma SH-SY5Y cells. Brain Res. 2002) in AD (Nizzari et al JAD 2012; Chun, Y.S., et al Mol Psychiatry 2022). On the other hands, the positive role of ERK in neuron survival, in memory formation as well as its inhibition in neurodegenerative diseases is well documented (Medina et al 2018; Bodai et al 2012; Melone et al 2013; Fusco FR et al 2012). It should be pointed out that a dual role of erk depends on the kinetics of activation and on its cellular compartmentalization (Colucci-d'amato 2003). Thus, the authors would provide a better signaling context for SAHA effects, highlighting that, according to the literature, it is likely that SAHA not only activates ERK but does it in a pro-survival mode.

R2) We thank the reviewer for this comment on the implication of ERK pathway.

Based on the suggestion of the reviewer, we have added a new comment in the revised MS Verrillo et al. R1:

Section 2.1 Neuronal maturation and plasticity, lane 130, page 4:

“This activity is extremely interesting, because in some instances ERK activation may lead to detrimental effects and contribute to neurodegeneration; or in other cases it fosters survival, differentiation or counteracts pathogenic effects of mutations in Huntington’s disease models (HD).

Indeed, several studies have reported the dual role of ERK1/2 signaling, as a coin with a double face [Colucci-D’amato 2003], that in response to specific signals may lead to beneficial effects on long-term and short-term memory formation and neuronal protective activity in Huntington’s disease models (HD) [Medina and Viola 2018; Fusco et al. 2012], but also detrimental effects inducing neurotoxicity [Nizzari et al. 2012]. This double role of ERK1/2 signaling may reflect differences into the functional response to SAHA, depending on target cells and tissue, as well as the dosage used [Chen et al. 2012; Kurundkar et al. 2013].

New references included:

Chen, S. H.; Wu, H. M.; Ossola, B.; Schendzielorz, N.; Wilson, B. C.; Chu, C. H.; Chen, S. L.; Wang, Q.; Zhang, D.; Qian, L.; Li, X.; Hong, J. S.; Lu, R. B. Suberoylanilide hydroxamic acid, a histone deacetylase inhibitor, protects dopaminergic neurons from neurotoxin-induced damage. Br. J. Pharmacol., 2012; 165:494–505.

Colucci-D'Amato L, Perrone-Capano C, di Porzio U. Chronic activation of ERK and neurodegenerative diseases. Bioessays. 2003 Nov;25(11):1085-95. doi: 10.1002/bies.10355. PMID: 14579249.

Kurundkar, D.; Srivastava, R. K.; Chaudhary, S. C.; Ballestas, M. E.; Kopelovich, L.; Elmets, C. A.; Athar, M. Vorinostat, an HDAC inhibitor attenuates epidermoid squamous cell carcinoma growth by dampening mTOR signaling pathway in a human xenograft murine model. Toxicol. Appl. Pharmacol. 2013; 266:233–244.

Medina JH, Viola H. ERK1/2: A Key Cellular Component for the Formation, Retrieval, Reconsolidation and Persistence of Memory. Front Mol Neurosci. 2018 Oct 5;11:361. doi: 10.3389/fnmol.2018.00361.

Fusco FR, Anzilotti S, Giampà C, Dato C, Laurenti D, Leuti A, Colucci D'Amato L, Perrone L, Bernardi G, Melone MA. Changes in the expression of extracellular regulated kinase (ERK 1/2) in the R6/2 mouse model of Huntington's disease after phosphodiesterase IV inhibition. Neurobiol Dis. 2012 Apr;46(1):225-33.

Nizzari M, Venezia V, Repetto E, Caorsi V, Magrassi R, Gagliani MC, Carlo P, Florio T, Schettini G, Tacchetti C, Russo T, Diaspro A, Russo C. Amyloid precursor protein and Presenilin1 interact with the adaptor GRB2 and modulate ERK 1,2 signaling. J Biol Chem. 2007 May 4;282(18):13833-44.

Q3) In the section, "SAHA in other neurological diseases" the authors overlooked an important effect of SAHA, that is the ability to inhibit vasculogenic mimicry in GBM, likely by altering neural transdifferention into endothelial-like cells. This has to do with plasticity thus it is pertinent to the review. It is worth to add this part in the text as well as in Fig.2 (biological effects: vasculogenic mimicry) and in Table 2 (Human disease: GBM)

R3) We thank the reviewer for this comment which we feel is beyond the original scope of the review, but nevertheless provides an opportunity for discussion.

As extensively reported along the manuscript, the focus of our review is on the applicability of SAHA as neuroplasticity drug in neurological diseases, excluding tumors. However, the Reviewer highlighted an interesting study describing how SAHA inhibits vasculogenic mimicry in glioblastoma multiforme cell lines, perhaps by altering neural transdifferention into endothelial-like cells.

As extensively reported in the literature, vasculogenic mimicry is caused by the formation of a “vessel-like” structure without endothelial cells in vascular-dependent solid tumors and it represents a special blood supply source involved in the highly invasive tumor progression. This formation has been identified in variety of tumors as uveal, cutaneous and mucous membrane melanomas, inflammatory and ductal breast carcinoma, ovarian and prostatic carcinoma, and soft tissue sarcomas, including synovial sarcoma rhabdomyosarcoma, osteosarcoma, pheochromocytoma.  In glioma, these vessel-like” structures are related to the activity of different molecular pathways, including hypoxia-related signaling pathways, matrix metalloproteinases, non-coding RNA activity, etc (Cai et al. 2020). To date how vasculogenic mimicry can impact on neuronal plasticity or other functions of neurons is unknown. In addition, very few studies carried out in glioma cell lines have described a potential involvement of astrocytes in vasculogenic mimicry (VM) formation (Liang et al. 2021-2022; Zhang et al. 2016; Liu et al. 2011). In a recent study Pastorino et al 2019 report that in human glioblastoma cell lines SAHA acts as anti-angiogenic drug. Although the molecular mechanism of action remains to be elucidated, the authors hypothesize that similarly to another HDACi, Trichostatin A (TSA) [Svechnikova et al. 2008], SAHA could interfere with the differentiation status of GBM cell lines by altering the expression of neuronal markers and consequently triggering the transition from epithelial-derived cells to mesenchymal cells.

Based on these considerations, in order to enrich the description about the huge versatility of SAHA and add a novelty about a potential new mechanism of action of SAHA, we have added the following sentence in the revised MS Verrillo et al. R1:

Section 4. SAHA in other neurological disorders, lane 464, page 13:

“Recently, in human glioblastoma cell lines, it has been observed a peculiar anti-angiogenic property of SAHA, capable to inhibit vasculogenic mimicry by interfering with the differentiation status of GBM cells [Pastorino et al 2019], perhaps by altering expression of neuronal markers that in turn can trigger the epithelial-mesenchymal transition, as also observed in GBM cell lines exposed at low concentrations of the HDACi Trichostatin A (TSA) [Svechnikova et al. 2008].

New references included:

Svechnikova I, Almqvist PM, Ekström TJ. HDAC inhibitors effectively induce cell type-specific differentiation in human glioblastoma cell lines of different origin. Int J Oncol. 2008 Apr;32(4):821-7. PMID: 18360709.

Pastorino O, Gentile MT, Mancini A, Del Gaudio N, Di Costanzo A, Bajetto A, Franco P, Altucci L, Florio T, Stoppelli MP, Colucci-D'Amato L. Histone Deacetylase Inhibitors Impair Vasculogenic Mimicry from Glioblastoma Cells. Cancers (Basel). 2019 May 29;11(6):747. doi: 10.3390/cancers11060747.

Q4) In the Introduction: from a lane 28 (Neuroplasticity...) to lane 35 (synaptic network…). A reference should be provided.

R4) Thank you for the note, we have added the new reference:

von Bernhardi, R., Bernhardi, L. E., & Eugenín, J. (2017). What Is Neural Plasticity? Advances in experimental medicine and biology1015, 1–15.

Q5) In Concluding Remarks: lanes 384-386: "The combination ....overcoming contradictory information reported in the previous literature ". examples should be given

R5) Many thanks for this note. As underlined previously, SAHA response can vary depending on target cells and tissue as well as the dosage used (see Reviewer 2 Q1). We also add some examples about the positive effects on neurothrophine levels (e.g. Bdnf) or the reduction of oxidative stress [Misztak 2021, Athira 2018; Ershadi 2021; Chen 2019].

Here the new sentences included in the revised MS Verrillo et al. R1:

  1. Concluding Remarks, lane 474, page 13:

“…..about its neuroprotective or neurotoxic effects in CNS that may depend on target cells and tissues, as well as the dosage used [Chen et al. 2012; Kurundkar et al. 2013]. Indeed, ….. for example by increasing neurothrophine levels (e.g. Bdnf) or counteracting oxidative stress [Misztak 2021, Athira 2018; Ershadi 2021; Chen 2019].

New references included:

Chen, S. H.; Wu, H. M.; Ossola, B.; Schendzielorz, N.; Wilson, B. C.; Chu, C. H.; Chen, S. L.; Wang, Q.; Zhang, D.; Qian, L.; Li, X.; Hong, J. S.; Lu, R. B. Suberoylanilide hydroxamic acid, a histone deacetylase inhibitor, protects dopaminergic neurons from neurotoxin-induced damage. Br. J. Pharmacol., 2012; 165:494–505.

Kurundkar, D.; Srivastava, R. K.; Chaudhary, S. C.; Ballestas, M. E.; Kopelovich, L.; Elmets, C. A.; Athar, M. Vorinostat, an HDAC inhibitor attenuates epidermoid squamous cell carcinoma growth by dampening mTOR signaling pathway in a human xenograft murine model. Toxicol. Appl. Pharmacol. 2013; 266:233–244.

Minor points:

M1. legend to fig. 1: line 46 the article is missing

R1. Thank you. The change was done at page 2 line 47 in the revised version of the MS.

M2. pag 2 line 56 the abbreviation ID has been cited for the first time but there is no explanation.

R2. Thank you. The change was done at page 2 line 54 in the revised MS Verrillo et al. R1:

M3. pag3 line 68 ...."in red": I should be capital letter

R3. Thank you. The change was done at page 3 line 96 in the revised MS Verrillo et al. R1:

M4. in authors contributions the name of Alberto de Bellis, one of the coauthor is absent.

R4. Thank you. We added the abbreviation of the co-authors Alberto de Bellis and also of Rosita Di Palma.

Round 2

Reviewer 1 Report

Verrillo et al. reviewed suberoylanilide hydroxamic acid (SAHA) as a useful agent to improve memory, learning, and behavior and to correct deficits in neuronal function. Their revised manuscript has improved well in accordance with the reviewer’s comments.  Therefore, this revised manuscript is considered as acceptable for the publication in “Biomolecules”.